# Long Non-Coding RNA GAS5 Promotes BAX Expression by Competing with microRNA-128-3p in Response to 5-Fluorouracil

**DOI:** 10.3390/biomedicines11010058

**Published:** 2022-12-26

**Authors:** Heejin Lee, Hoin Kang, Chongtae Kim, Ja-Lok Ku, Sukwoo Nam, Eun Kyung Lee

**Affiliations:** 1Department of Biochemistry, College of Medicine, The Catholic University of Korea, Seoul 06591, Republic of Korea; 2Department of Internal Medicine, College of Medicine, The Catholic University of Korea, Seoul 06591, Republic of Korea; 3Department of Biomedical Sciences, College of Medicine, Seoul National University, Seoul 03080, Republic of Korea; 4Department of Pathology, College of Medicine, The Catholic University of Korea, Seoul 06591, Republic of Korea; 5Institute for Aging and Metabolic Diseases, College of Medicine, The Catholic University of Korea, Seoul 06591, Republic of Korea

**Keywords:** long non-coding RNA, GAS5, competing endogenous RNA, BAX, microRNA

## Abstract

The acquisition of drug resistance is a major hurdle for effective cancer treatment. Although several efforts have been made to overcome drug resistance, the underlying mechanisms have not been fully elucidated. This study investigated the role of long non-coding RNA (lncRNA) growth arrest-specific 5 (GAS5) in drug resistance. GAS5 was found to be downregulated in colon cancer cell lines that are resistant to 5-fluorouracil (5-FU). Downregulation of GAS5 decreased the viability of HCT116 cells and the level of the pro-apoptotic BAX protein, while GAS5 overexpression promoted cell death in response to 5-FU. The interaction between GAS5 and *BAX* mRNA was investigated using MS2-tagged RNA affinity purification (MS2-trap) followed by RT-qPCR, and the results showed that GAS5 bound to the 3′-untranslated region of *BAX* mRNA and enhanced its expression by interfering with the inhibitory effect of microRNA-128-3p, a negative regulator of BAX. In addition, ectopic expression of GAS5 increased the sensitivity of resistant cells in response to anti-cancer drugs. These results suggest that GAS5 promoted cell death by interfering with miR-128-3p-mediated BAX downregulation. Therefore, GAS5 overexpression in chemo-resistant cancer cells may be a potential strategy to improve the anti-cancer efficacy of drugs.

## 1. Introduction

Resistance to anti-cancer drugs is a major impediment that leads to cancer treatment failure [1,2,3,4]. Although many types of cancer are initially susceptible to chemotherapy, some cells can develop resistance to anti-cancer drugs that can contribute to the recurrence of cancers. Cancer cells become resistant to anti-cancer drugs through various mechanisms, including drug inactivation, inhibition of cell death, changes in drug metabolism, enhanced DNA repair, amplification of target genes, and increased proliferation [5,6]. 5-Fluorouracil (5-FU), a commonly used anti-cancer drug in the treatment of colon cancer, is a synthetic fluorinated pyrimidine analog that causes cell death by interfering with nucleoside metabolism [7,8]. Despite the effective chemotherapeutic role of 5-FU, its clinical applications are largely limited due to drug resistance. Several efforts have been made to define the key determinants of drug resistance; however, the detailed mechanism of 5-FU-induced drug resistance remains largely unknown [4,9,10].

Recent studies showing genome-wide transcriptome analyses have revealed the differential expression of various types of non-coding RNAs (ncRNAs), including microRNAs (miRNAs), circular RNAs (circRNAs), and long ncRNAs (lncRNAs), and their critical roles in the regulation of drug resistance [11,12,13,14]. LncRNAs are a type of ncRNA longer than 200 nucleotides that do not synthesize protein and function as an essential regulator of gene expression at multiple levels by interacting with DNA, RNA, and proteins [15,16,17]. They affect cell growth, survival, death, differentiation, and response to diverse stimuli, and their abnormal expression is related to various pathophysiological processes [15,18,19]. LncRNAs function as ‘competing endogenous RNAs (ceRNAs)’ by impairing miRNA activity through sequestration, therefore, upregulating the expression of miRNA target genes [20,21,22,23]. LncRNA growth arrest-specific 5 (GAS5) is important in regulating cell growth, differentiation, and development [24,25]. The aberrant expression of GAS5 is associated with several diseases, including various cancers, neurological disorders, and bone diseases [25,26,27]. Several studies have reported tumor suppressive roles of GAS5, and its reduction in malignant tumors is associated with an increase in malignancy, a poor prognosis, and drug resistance [25,27]. As a ceRNA, GAS5 interferes with the action of several miRNAs, including miR-21 [28], miR-23 [29], miR-32-5p [30], miR-128-3p [31,32], miR-196a-5p [33], miR-222-3p [34], and miR-378a-5p [35], thereby affecting cancer progression.

In this study, the role of GAS5 in 5-FU resistance was investigated in human colon cancer cells. Cells resistant to 5-FU had a lower level of GAS5, and ectopic expression of GAS5 increased the sensitivity of cells resistant to 5-FU by promoting cell death. A pull-down assay revealed that GAS5 interacted with *BAX* mRNA and promoted its expression by completing with miR-128-3p. Our results demonstrate the role of GAS5 as a ceRNA of miR-128-3p in 5-FU resistance by promoting BAX-inducing cell death and suggest a potential use of GAS5 for effective chemotherapy to improve the efficacy of anti-cancer drugs.

## 2. Materials and Methods

### 2.1. Cell Culture, Cloning, and Transfections

Human colon cancer cell lines, including SNU-C4, SNU-C5 (obtained from the Korean Cell Line Bank (Korea)), and HCT116 (obtained from the American Type Culture Collection (ATCC)), were maintained in Roswell Park Memorial Institute medium (RPMI-1640) (Thermo Fisher Scientific, Waltham, MA, USA) supplemented with 10% fetal bovine serum (FBS) and 1% penicillin/streptomycin (P/S). Cells resistant to 5-FU (SNU-C4R and SNU-C5R obtained from the Korean Cell Line Bank [36] and HCT116R generated in a previous study [37]) were maintained in RPMI/FBS/P/S media containing IC_50_ values of 5-FU. The GAS5 overexpression plasmid (pGAS5) was constructed by inserting the GAS5 gene (NR_002574.2) into pcDNA3 (Thermo Fisher Scientific). The reporter plasmid was generated by inserting the 3′UTR of *BAX* mRNA into pEGFP-C1 (Clontech, Takara Bio, Shiga, Japan). Transfection of siRNAs (siCtrl, 5′-AAUUCUCCGAACGUGUCACGU-3; siGAS5, 5-CUGAAGUCCUAAAGAGCAAUU-3) (Genolution, Seoul, Korea), miRNAs (Bioneer, Daejeon, Korea), and plasmids were performed using Lipofectamine 2000 (Thermo Fisher Scientific), according to the manufacturer’s instructions.

### 2.2. RNA Analysis and RT-qPCR

Total RNAs were isolated from cells using RNAiso Plus (Takara Bio, Inc., Shiga, Japan), and complementary DNA (cDNA) was synthesized by reverse transcription (RT) using the ReverTra Ace™ qPCR RT Kit (Toyobo Co., Ltd., Osaka, Japan). The abundance of transcripts was assessed through RT-qPCR analysis using the SensiFAST™ SYBR Hi-ROX kit (Meridian Bioscience, Inc., Cincinnati, OH, USA) and gene-specific primers listed in Appendix A. *GAPDH* mRNA was used as an internal control for the normalization of gene expression measurements. Data were processed using the _ΔΔ_CT method for comparison between control and experimental groups.

### 2.3. Western Blotting

Whole-cell lysates were prepared using RIPA buffer (10 mM Tris–HCl (pH 7.4), 150 mM NaCl, 1% NP-40, 1 mM EDTA, 0.1% sodium dodecyl sulfate) containing 1× protease inhibitor cocktail. The samples were separated by SDS-PAGE, transferred onto polyvinylidene difluoride (PVDF) membranes (Millipore, Burlington, MA, USA), and incubated with primary antibodies against cleaved caspase 3 (c-Casp3) (Cell Signaling Technology, Danvers, CA, USA), cleaved poly-ADP-ribose polymerase (c-PARP) (Abcam, Cambridge, MA, USA), GFP (Santa Cruz Biotech, Santa Cruz, CA, USA), BAX (Santa Cruz Biotech), and β-actin (Abcam). The membranes were incubated with the appropriate secondary antibodies conjugated to horseradish peroxidase (HRP) (Sigma-Aldrich Inc., St. Louis, MO, USA), and chemiluminescence was detected with the Clarity Western ECL Substrate (Bio-Rad Laboratories, Inc., Hercules, CA, USA).

### 2.4. Cell Viability, Colony Forming Assay, and Flow Cytometric Analysis

The MTT assay was used to determine cell viability by adding 0.5 mg/mL of the 3-(4, 5-dimethylthiazol-2-yl)-2,5-diphenyltetrazolium bromide solution into the cells. The medium was removed after 3 h incubation, and the crystal was solubilized with 100 microliter of 40 mM acidic isopropanol. The optical absorbance of the solution was measured at 570 nm using a Victor 3 microplate reader (Perkin Elmer, Turku, Finland). For colony forming assay, cells were split into 1 × 10^3^ cells per well and incubated further with anticancer drugs. After two weeks, the cells were fixed with 4% paraformaldehyde and stained with 0.05% crystal violet solution. After washing and drying the stained plates, the number and size of colonies were assessed using the Image J software from three random microscopic field images per sample. For the flow cytometric analysis, cells were incubated with propidium iodide (Sigma-Aldrich, Burlington, MA, USA), and cell cycle distribution was analyzed by flow cytometry using the BD FACSCalibur™ (BD Bioscience, Farmingdale, NY, USA). The cells in the sub G1 portion were counted, and the relative fold changes were analyzed using BD CellQuest™ Pro 5.2.1 (BD Bioscience) and WinMDI 2.9 software.

### 2.5. MS2-Trap Assay and Biotin Pull-Down Assay

To generate a stable cell line expressing MS2 binding protein (MS2BP), Hep3B cells were transfected with pFlag-MS2BP and exposed to 500 μg/mL of Geneticin (Invitrogen) for two weeks. The clones which survived were isolated and expanded as the Hep3B/F-MS2BP cells. For the analysis of the interaction between GAS5 and its associating RNAs, the Hep3B/F-MS2BP cells were transfected with the plasmid containing tandem repeats of MS2 hairpin RNAs and GAS5 (pGAS5), and then the complex of MS2 RNAs and F-MS2BP was pulled down by immunoprecipitation using anti-Flag antibody [38]. For the biotin pull-down assay, the GAS5 gene was amplified using the forward primer listed in Appendix A, and biotinylated GAS5 was synthesized by in vitro transcription using the MaxiScript T7 kit (Invitrogen™) and biotin-CTP (Enzo Life Sciences, Inc., Farmingdale, NY, USA). The complexes containing GAS5 and its interacting RNAs were isolated using streptavidin-coupled Dynabeads [39]. The RNAs in the complex were isolated, transcribed, and analyzed by RT-qPCR. The 3′UTR of *GAPDH* mRNA was used as a control RNA for the binding assay.

### 2.6. Statistical Analysis

Data was expressed as mean ± SEM of three independent experiments. The statistical significance of the data was analyzed via Student’s *t*-test (*, *p* < 0.05).

## 3. Results

Our previous study using human colon cancer cells (SNU-C4 and SNU-C5) and their 5-FU resistant cells (SNU-C4R and SNU-C5R) revealed lncRNA GAS5 as one of the downregulated lncRNAs in the resistant cells [40]. GAS5 levels in both SNU-C4R and SNU-C5R cells were significantly reduced when compared to their parental cells (Figure 1). Reduction of GAS5 was also observed in another 5-FU resistant cell line which is generated from HCT116 cell (HCT116R) [37] (Figure 1). These results suggest that the levels of GAS5 are reduced in 5-FU resistant colon cancer cells.

To determine whether GAS5 downregulation is associated with 5-FU response, HCT116 cells transiently transfected with siRNAs or plasmids were exposed to 5-FU and then analyzed for cell viability, growth, and death via MTT assay, colony-forming assay, flow cytometry, and western blotting analysis. GAS5 knockdown increased cell viability and the number of colonies in response to 5-FU, while GAS5 overexpression decreased them (Figure 2A,B). In addition, GAS5 knockdown reduced the cell populations in the sub G1 phase and the level of apoptotic marker proteins, including cleaved caspase-3 and cleaved PARP, after 5-FU treatment (Figure 2C,D). In contrast, GAS5 overexpression promoted 5-FU-induced cell death. These results suggest the potential role of GAS5 in regulating 5-FU resistance in colon cancer cells.

The MS2-trap assay was performed to investigate the interaction between GAS5 and its molecular targets by generating two plasmids, pMS2-GAS5 and pFlag-MS2BP, as shown in Figure 3A. pMS2-GAS5 and pFlag-MS2BP were designed to produce the chimeric RNA of the MS2 hairpin RNAs-GAS5 and the Flag-tagged MS2BP protein, respectively. After establishing a stable cell line expressing Flag-MS2BP (Hep3B/F-MS2BP) (Figure 3A, right), the cells were transfected with pMS2 or pMS2-GAS5 and then the ribonucleoprotein (RNP) complex of MS2-GAS5 and MS2BP was immunoprecipitated using Flag antibody. Pulldown of GAS5 was assessed by RT-qPCR using the isolated RNAs in the RNP complex (Figure 3B, middle).

Based on our observation showing the promotion of cell death by GAS5 in response to 5-FU exposure (Figure 2) and a recent study by Wang and colleagues [31], we evaluated the pro-apoptotic protein BAX as a target of GAS5. The enrichment of *BAX* mRNA in the GAS5-containing RNP complex was analyzed by RT-qPCR, and the results showed that *BAX* mRNA was moderately, but significantly, enriched in the GAS5-containing RNP complex (Figure 3B, right), which suggests an association between GAS5 and *BAX* mRNA. The binding of GAS5 to *BAX* mRNA was further investigated by biotin pulldown assay and in silico alignment using the BLAST algorithm (https://blast.ncbi.nlm.nih.gov/Blast.cgi, accessed on 15 November 2022). As shown in Figure 3C, biotinylated GAS5 pulled down *BAX* mRNA, while a control *GAPDH* mRNA did not. A survey using the BLAST algorithm found the potential regions of base-paring between *BAX* mRNA and GAS5 in its 3′UTR regions (563–574 nt and 895–909 nt) (Figure 3D). These results suggest that *BAX* mRNA is one of the binding partners of GAS5.

To determine whether GAS5 regulates BAX expression, HCT116 cells were transfected with siRNAs or plasmids, and then BAX expression was assessed by RT-qPCR and western blotting analysis after 5-FU treatment. The levels of *BAX* mRNA were not significantly changed by GAS5 regulation. However, GAS5 positively regulated BAX protein expression in HCT116 cells; the BAX protein level was decreased by GAS5 knockdown while increased by GAS5 overexpression in response to 5-FU (Figure 4A). GAS5-mediated BAX regulation was further analyzed by EGFP reporter assay. The reporter plasmid pEGFP-BAX 3U was generated by inserting the 563–909 nt region of *BAX* mRNA into pEGFP (Figure 4B), and relative EGFP level was assessed by western blotting analysis in HCT116 and HCT116R cell. As expected, GAS5 positively regulated the expression of EGFP reporters; the EGFP level was decreased by GAS5 knockdown while increased by GAS5 overexpression (Figure 4C). Taken together, these results indicate that GAS5 associates with *BAX* mRNA and positively regulates its expression. Since GAS5 decreased in 5-FU resistant cells (Figure 1) and regulated BAX expression (Figure 4A,C), relative BAX levels between 5-FU resistant cells and their parental cells were investigated by western blotting analysis. The result showed that 5-FU resistant cells had lower BAX expression than their parental cells (Figure 4D). These data suggest a positive correlation between GAS5 and BAX and its potential role in 5-FU resistance.

To address the possible mechanism of GAS5-mediated BAX regulation as a ceRNA, we examined the effect of miR-128-3p, one of the BAX-targeting miRNAs, on BAX expression [31,41,42]. HCT116 cells were co-transfected with GAS5 and miR-128-3p, and relative BAX expression was assessed by western blotting. As shown in Figure 5A, miR-128-3p suppressed BAX expression, whereas GAS5 overexpression overcame miR-128-3p-mediated BAX downregulation. In addition, the binding of GAS5 to *BAX* mRNA was determined by MS2-trap assay and biotin pulldown assay with or without miR-128-3p. Co-transfection of miR-128-3p reduced the binding of GAS5 to *BAX* mRNA, which indicates that miR-128-3p interferes with the association between GAS5 and *BAX* mRNA (Figure 5B,C). These results suggest that GAS5 enhances BAX expression by competing with miR-128-3p.

To determine whether GAS5 increases the effects of 5-FU in the resistant cells, HCT116R cells were transfected with pGAS5 and exposed to 5-FU. GAS5 overexpression decreased cell viability and the number of colonies in HCT116R cells (Figure 6A,B), which indicates that GAS5 increases the sensitivity of HCT116R cells to 5-FU treatment. Additionally, ectopic expression of GAS5 reduced the viability of HCT116R cells after treatment with various anti-cancer reagents such as doxorubicin (DOX), oxaliplatin (OXP), cisplatin (CDDP), and tamoxifen (TAM) (Figure 6C), which implies that GAS5 helps to improve drug response in resistant cells. Our findings suggest that GAS5 plays a key role in the regulation of anti-cancer drug resistance in colon cancer and has therapeutic potential for effective chemotherapy.

## 4. Discussion

LncRNA GAS5 was originally identified as part of a group of genes expressed during the growth arrest phase of NIH3T3 cells and has been characterized as a tumor suppressor [20]. Dysregulation of GAS5 is involved in the pathophysiological process of diseases, including various types of cancer, neurological disorders, and bone diseases [25,26]. A series of studies have shown that GAS5 plays a tumor-suppressive role by regulating the growth, death, migration, and invasion of cancer cells (reviewed in [27]). As a ceRNA, GAS5 interferes with the action of several miRNAs, including miR-21 [28], miR-23 [29], miR-32-5p [30], miR-128-3p [31,32], miR-196a-5p [33], miR-223-3p [34], and miR-378a-5p [35], and inhibits the ability of miRNAs to regulate their target genes, that can lead to cancer progression. Here, we demonstrate the role of GAS5 in the regulation of chemoresistance against an anti-cancer drug, 5-FU. Our results indicate that GAS5 promoted cell death and growth of HCT116 cells by enhancing the expression of pro-apoptotic protein BAX via the competitive interaction with miR-128-3p, thereby increasing the sensitivity of HCT116 cells in response to 5-FU. Recent studies by Wang [31] and Peng [32] reported the role of GAS5 in hypoxic-ischemic brain damage and rheumatoid arthritis progression via the miR-128-3p/BAX axis and miR-128-3p/HDAC4 axis. In this study, we also demonstrated the competitive relation between GAS5 and miR-128-3p on BAX regulation and its contribution as a ceRNA in anti-cancer drug resistance of colon cancer cells. The interaction between GAS5 and *BAX* mRNA was experimentally assessed by MS2-trap assay followed by RT-qPCR (Figure 3). The binding of *BAX* mRNA to GAS5 was evaluated based on cell death-related phenotype in our study system. The MS2-trap assay is an attractive method to study the interactions between lncRNAs and various binding partners, such as proteins, miRNAs, lncRNAs, and metabolites. Therefore, additional research using the MS2-trap assay may make it possible to elucidate the molecular targets of GAS5 and the detailed mechanisms by which GAS5-mediated biological function.

Several reports have shown that the relative expression of GAS5 decreases in various types of cancer, such as cervical cancer [43], breast cancer [44], and esophageal squamous cell carcinoma [45], and also in the cisplatin-resistant lung cancer A549 cells [46]. Furthermore, we have previously observed the reduction of GAS5 in 5-FU resistant colon cancer cells [40]. As shown in Figure 1, the relative expression of GAS5 was significantly lower in three different resistant colon cancer cell lines, including SNU-C4R, SNU-C5R, and HCT116R, compared to their parental cells. Alterations of mTOR signaling [47,48] or aberrant methylation of the GAS5 promoter region [49] have been suggested as factors affecting the cellular GAS5 level; however, the detailed mechanisms of how GAS5 expression is downregulated have not been fully elucidated. No significant change in GAS5 levels was observed in HCT116 cells in response to short-term exposure to 5-FU (data not shown), suggesting that complex alterations in gene expression during 5-FU exposure contribute to decreasing GAS5 levels. Future studies are required to fully elucidate the mechanism of how resistant cells have a lower level of GAS5, which leads to chemoresistance to anti-cancer drugs.

GAS5 overexpression increased the cell populations in the sub G1 phase while reducing the number of colonies after 5-FU treatment (Figure 2). In addition, ectopic expression of GAS5 promoted cell death in the resistant HCT116R cells in response to anti-cancer drugs (Figure 6), which implies the potential role of GAS5 in the regulation of chemosensitivity. Increased GR activity has been shown to promote the transcription of target genes containing glucocorticoid receptor elements (GREs) in colon cancer cells, contributing to cancer progression by increasing cancer cell metastasis [50]. Kino and colleagues recently reported that the lncRNA GAS5 suppresses the transcriptional activity of glucocorticoid receptor (GR) by interacting with the DNA binding domain of GR [51]. The GRE-mimic sequence of GAS5 functions as a regulatory repressor element of the GR, reducing the transcriptional activity of GR [51,52,53]. We did not investigate the GR activity in this study; however, the determination of whether the GR activity is upregulated in the 5-FU resistant cells would be a valuable challenge to understanding the molecular basis of GAS5-mediated cancer development.

Although the detailed mechanisms of GAS5-mediated regulation of drug response need to be further investigated, these findings suggest a potential strategy for combating anti-cancer drug resistance by targeting GAS5. GAS5 overexpression may be helpful to combat metastasis or drug response of drug-resistant colon cancer cells. Moreover, the relative level of GAS5 may be used as a useful prognostic marker to predict drug response or malignancy in cancer patients. Additional research using patient-derived tissues may provide a better understanding of the clinical significance of GAS5 and assess its therapeutic potential to improve chemotherapy efficacy.

Previous reports have shown that miRNA-128-3p suppresses BAX expression by targeting the 3′UTR of *BAX* mRNA [41], and the miR-128-3p/BAX axis is involved in the regulation of the chemosensitivity of breast cancer cells [42]. We demonstrate that lncRNA GAS5 enhances BAX expression by interfering with miR-128-3p, thereby contributing to decreasing resistance of colon cancer cells in response to anti-cancer drugs. miR-128-3p is known to regulate drug resistance by targeting various genes, including c-Met, MUC1-C, BMI-1, and ABCC5 in hepatocellular carcinoma [54], glioma [55], lung cancer [56], and ovarian cancer [57]. In this study, we did not investigate whether GAS5 binds to those target genes of miR-128-3p and influences their expression. Further study will provide experimental evidence showing the molecular mechanisms of GAS5 and its clinical significance in drug resistance.

In conclusion, the present study demonstrates that lncRNA GAS5 plays a tumor-suppressive role in colon cancer cells. The level of GAS5 decreased in 5-FU resistant cells, and its ectopic expression promoted cell death in response to anti-cancer drugs. Furthermore, as a ceRNA, GAS5 binds to *BAX* mRNA and interferes with miR-128-3p activity in BAX regulation. These findings suggest that GAS5 may be a promising target for overcoming chemoresistance to 5-FU in colon cancer.

## Figures and Tables

**Figure 1 biomedicines-11-00058-f001:**
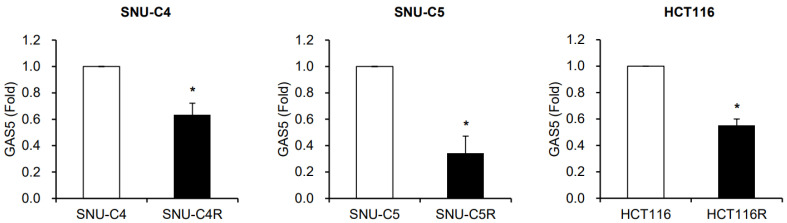
Downregulation of GAS5 in 5-FU resistant colon cancer cells (SNU-C4R, SNU-C5R and HCT116R) compared to their parental cells (SNU-C4, SNU-C5, and HCT116). Relative levels of GAS5 were determined between parental cells and resistant cells by RT-qPCR. *GAPDH* mRNA level was used for normalization. Data were expressed as the mean ± SEM from three independent experiments. The statistical significance of the data was analyzed via Student’s *t*-test; *, *p* < 0.05.

**Figure 2 biomedicines-11-00058-f002:**
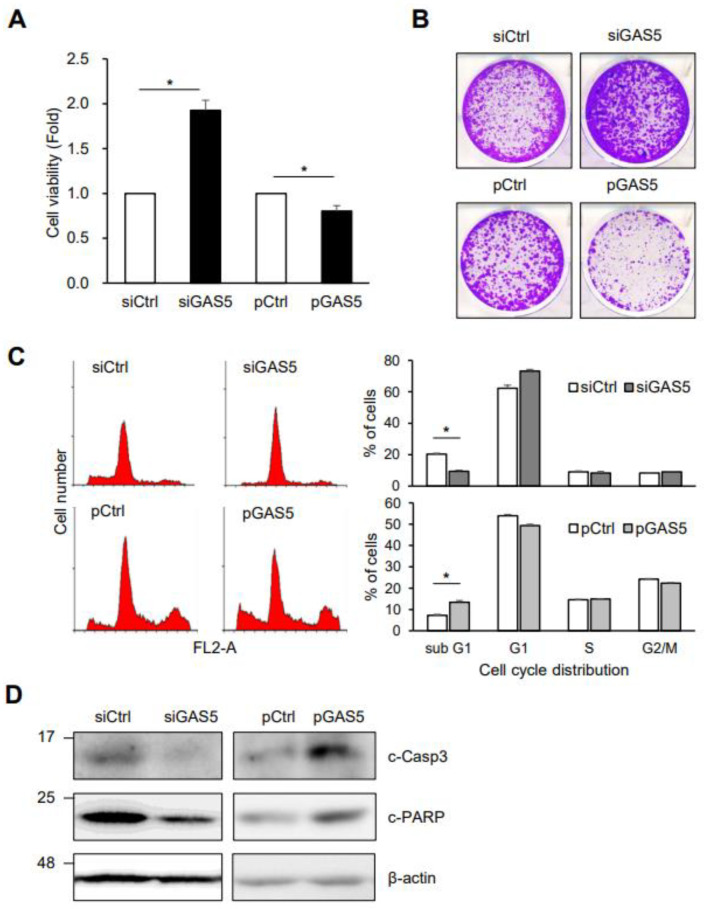
GAS5 promotes cell death in response to 5-FU. HCT116 cells were transfected with siRNAs or plasmids and further incubated with 5-FU. The effects of GAS5 on cell viability, growth, and death were assessed by MTT assay (**A**), colony-forming assay (**B**), flow cytometry (**C**), and western blotting analysis (**D**). Cell viability was determined by measuring the absorbance of the formazan crystals generated by cells. Colony-forming ability was assessed by counting the number of colonies after two weeks of incubation. Cells were stained with propidium iodide, and the distribution of the cell cycle and the cell populations in the sub G1 phase were analyzed by flow cytometry. For measuring apoptotic cell death, the levels of cleaved caspase-3 (c-Casp3) and cleaved PARP (c-PARP) were determined by western blotting analysis. β-actin was used as a loading control. Data were expressed as the mean ± SEM, and images are representative of three independent experiments. The statistical significance of the data was analyzed via Student’s *t*-test; *, *p* < 0.05.

**Figure 3 biomedicines-11-00058-f003:**
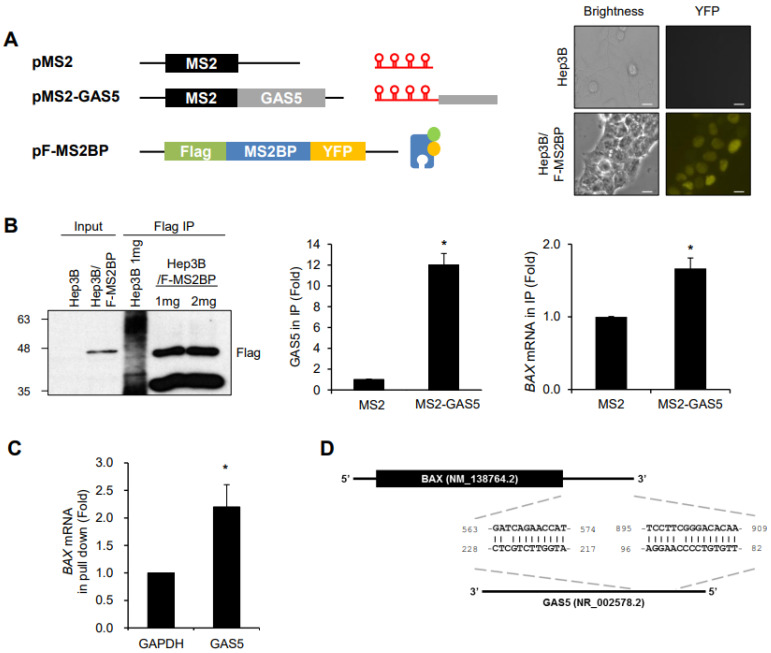
Association between GAS5 and *BAX* mRNA. (**A**) Schematics of the MS2-trap system. GAS5 was cloned behind MS2 hairpin sequences of pMS2 to generate pMS2-GAS5. A plasmid encoding Flag-tagged MS2BP-YFP (pFlag-MS2BP) was transfected with Hep3B cells to establish a stable cell line expressing Flag-MS2BP (Hep3B/F-MS2BP). The fluorescence of YPF was observed using a fluorescence microscope. Scale bar, 20 μm. (**B**) After transfection of pMS2 or pMS2-GAS5, the F-MS2BP containing ribonucleoprotein complex was isolated from the cell lysates using Flag Ab-coated beads, and the enrichment of F-MS2BP in the complex was shown by western blotting analysis (left). Relative levels of GAS5 and *BAX* mRNA in the complex were assessed by RT-qPCR (right). (**C**) Biotin pull-down assay. Biotin-labeled GAS5 was generated and incubated with HCT116 cell lysates. The GAS5-containing RNA complex was isolated using a Streptavidin bead, and the association between GAS5 and *BAX* mRNA in the complex was analyzed by RT-qPCR. *GAPDH* mRNA was used for the normalization. (**D**) In silico analysis of the interaction between GAS5 and *BAX* mRNA using the BLAST algorithm. Data were expressed as the mean ± SEM, and images are representative of three independent experiments. The statistical significance of the data was analyzed via Student’s *t*-test; *, *p* < 0.05.

**Figure 4 biomedicines-11-00058-f004:**
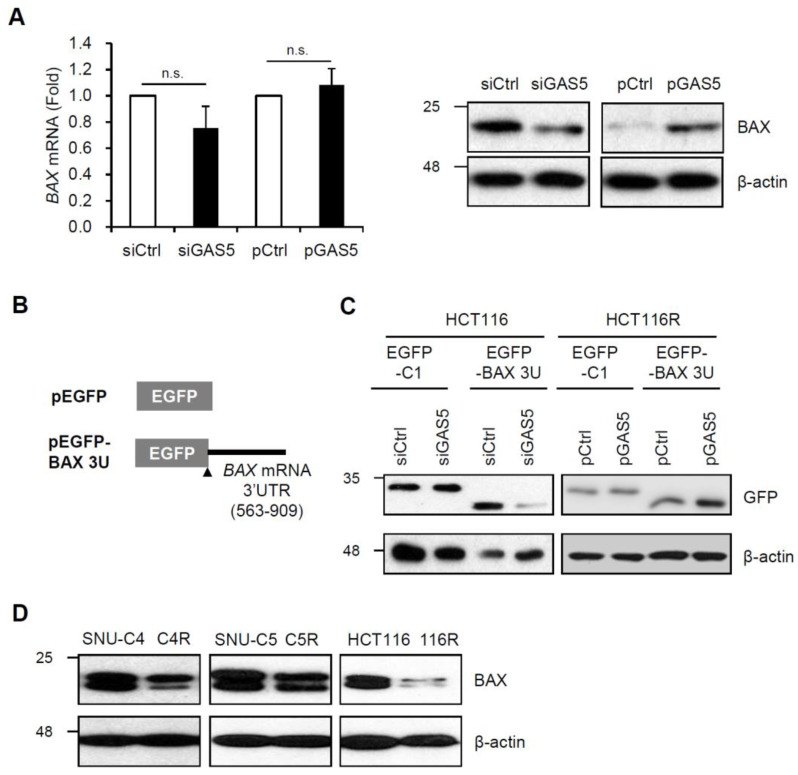
Regulation of BAX expression by GAS5. (**A**) HCT116 cells transfected with siRNAs or plasmids were incubated with 5-FU for 24 h. Relative BAX expression was assessed by RT-qPCR and western blotting analysis. *GAPDH* mRNA was used for the normalization. β-actin was used as a loading control. (**B**,**C**) The EGFP reporter assay. (**B**) Schematics of the EGFP reporters. The reporter plasmid containing the 3′UTR of *BAX* mRNA (pEGFP-BAX 3U) was generated by inserting the 563-909 nt region of *BAX* mRNA into pEGFP. (**C**) After sequential transfection of siRNAs or plasmids with the EGFP reporters, relative reporter expressions between HCT116 cells and HCT116R cells were determined by western blotting analysis. Data were expressed as mean ± SEM, and images are representative of three independent experiments. (**D**) Relative BAX expressions between 5-FU resistant cells and their parental cells were assessed by western blotting analysis. β-actin was used as a loading control. Images are representative of three independent experiments. The statistical significance of the data was analyzed via Student’s *t*-test; n.s., not significant.

**Figure 5 biomedicines-11-00058-f005:**
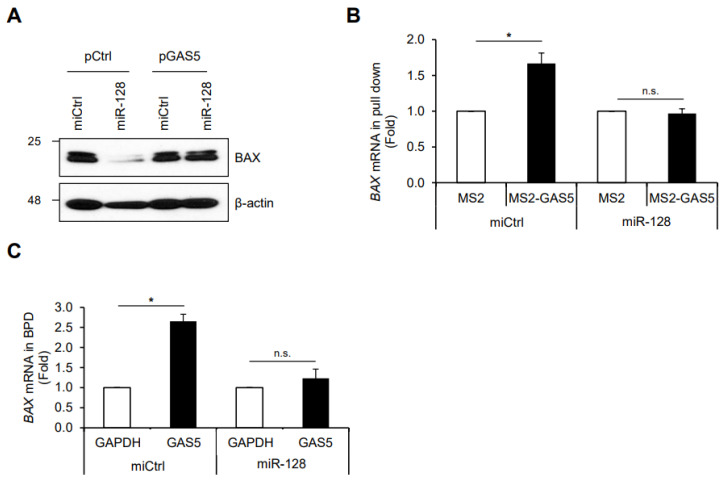
Competitive regulation of BAX expression by GAS5 and miR-128-3p. (**A**) After the co-transfection of plasmids and miRNAs into HCT116 cells, relative BAX expression was assessed by western blotting analysis. β-actin was used as a loading control. (**B**) Hep3B/F-MS2BP cells were co-transfected with plasmids and miRNAs; the MS2-GAS5-containing ribonucleoprotein complex was isolated using Flag-Ab beads. The enrichment of *BAX* mRNA in the complex was analyzed by RT-qPCR. (**C**) After transfection of HCT116 cells with miRNAs, cell lysates were incubated with biotin-labeled transcripts, and the GAS5-containing RNA complex was isolated using a Streptavidin bead. The association between GAS5 and *BAX* mRNA in the complex was analyzed by RT-qPCR. *GAPDH* mRNA was used for the normalization. Data were expressed as mean ± SEM, and images are representative of three independent experiments. The statistical significance of the data was analyzed via Student’s *t*-test; *, *p* < 0.05, n.s., not significant.

**Figure 6 biomedicines-11-00058-f006:**
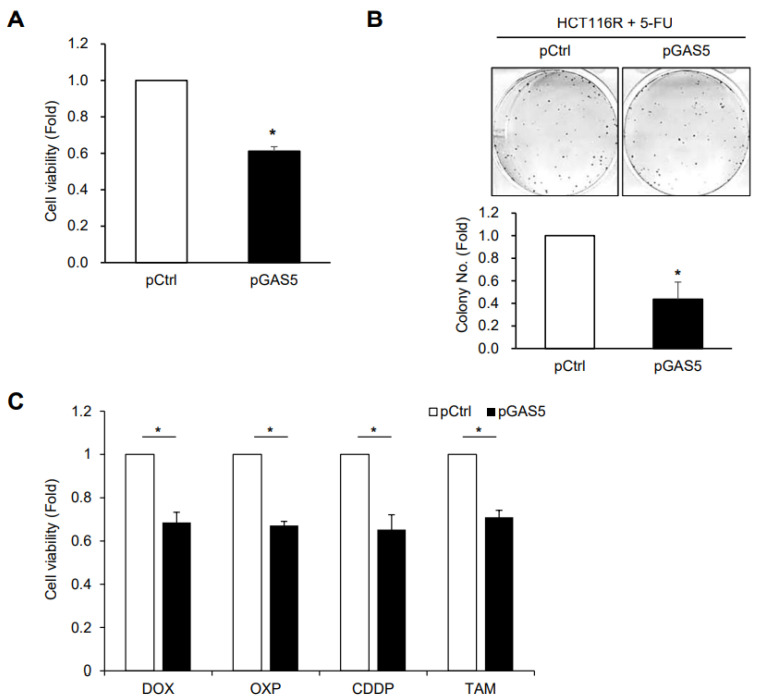
Sensitization of resistant cells by ectopic expression of GAS5. (**A**,**B**) HCT116R cells were transfected with plasmids and further incubated with 5-FU for 48 h. Cell viability was assessed by MTT assay (**A**) and colony-forming assay (**B**). (**C**) HCT116R cells transfected with plasmids were incubated with various anti-cancer drugs such as doxorubicin (DOX), oxaliplatin (OXP), cisplatin (CDDP), and tamoxifen (TAM) for 48 h, and cell viability was assessed by MTT assay. Data were expressed as mean ± SEM, and images are representative of three independent experiments. The statistical significance of the data was analyzed via Student’s *t*-test; *, *p* < 0.05.

## Data Availability

The data used to support the findings of this study are available from the corresponding author upon request.

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
