# Peer review of "Long Non-Coding RNA GAS5 Promotes BAX Expression by Competing with microRNA-128-3p in Response to 5-Fluorouracil"

_biomedicines, 2022, doi:10.3390/biomedicines11010058_

Round 1
Reviewer 1 Report
biomedicines-2068790
This study addresses the issue of drug resistance in tumor cells, which is of paramount importance for patient survival and in the clinic, and therefore the subject is of keen interest to the general readership of the Journal. The authors use human colorectal cancer cell lines resistant to 5-fluorouracil, a drug frequently used in the clinic, to address the role of GAS5, a long noncoding RNA gene, in drug resistance. By using a combination of methods, including cell culture, cloning, RNA interference, RT-qPCR, MS2-trap, pull-down and other assays, the authors report that resistance to 5-fluorouracil stems in part from a downregulation of GAS5, which limits synthesis of the proapoptotic tumor suppressor BAX by failing to interference with a microRNA, miR-128-3p, a BAX inhibitor. This conclusion appears well supported by the data. Overall, the manuscript is well written, data are clearly presented, figures are informative, and methods are described in detail.
I only have minor suggestions.
Abstract – “These results suggest that GAS5 promoted cell death by interfering with miR-128-3p-mediated BAX downregulation and restoring GAS5 levels in chemo-resistant cancer cells.” This sentence is unclear; how can GAS5 restore its own levels in chemo-resistant cancer cells? Isn’t GAS5 downregulated in chemo-resistant cells?
Page 2. “Some of lncRNAs” Unclear. – Please consider: Some lncRNAs function as competitors of endogenous RNAs (ceRNA), including impairing miRNA.
Page 3 – I would suggest using “mL” (capital L for milliliter) consistently throughout the manuscript.
Page 4 – “Our previous study … discovered lncRNA” – maybe uncovered, revealed?
Page 4 – “These results suggest that the levels of reduction of GAS5 are reduced in 5-FU resistant colon cancer cells” – Please consider: These results suggest that the levels of GAS5 are reduced in 5-FU resistant colon cancer cells.
Figure 2, legend – It may be useful to remind the reader that pGAS5 overproduces GAS5. How about this title: “GAS5 promotes cell death in response to 5-FU”.
Figure 2 – It may be useful to add 2 lines above the bars in A and C to specify which pairs are being compared. Same applies to other figures as well.
Figure 3, legend – “Scale bar,20 …” please add a space after the comma. “are representative from three independent” , are representative of three…Please check other legends as well.
Page 7 – “The levels of BAX mRNA were not significantly changed by GAS5 regulation, however, GAS5 positively regulated BAX expression in HCT116 cells” - This sentence is not clear.
Figure 4A and elsewhere – please indicate if the differences are not significant.
Page 7 – “the EGFP level was decreased by GAS5 knockdown while increased by GAS5 overexpression (Fig. 4B)”. Is it 4B or 4A?
Page 8 – “whereas GAS5 overexpression restored” – maybe overcame or abolished instead of “restored”.
Page 9 – “LncRNA GAS5 was originally identified as a group of genes expressed” – do you mean that LncRNA GAS5 was originally identified as part of a group of genes expressed …?
Page 10 – “known to regulate drug resistance by targeting various targets” – maybe targeting various genes?
Reviewer 2 Report
In the present study, the authors investigated the role of GAS5 in 5-fluorouracil (5-FU) resistance, in human colon cancer cells. They reported that cells resistant to 5-FU had a lower level of GAS5 and ectopic expression of GAS5 increased the sensitivity of cells resistant to 5-FU by promoting cell death.
Their work is very interesting, as they have performed a plethora of experimental work, in order to support their hypothesis. Their findings are enforcing the role of GAS5 in tumors.
Their manuscript has merit for publication after addressing some minor issues.
Since GAS5 was found to bind to the glucocorticoid receptor, how is can this be linked to the findings of the present study? The authors should mention the role of GAS5 and its relation to GR (PMID: 27214311, PMID: 20124551), as well as the role of GAS5 in tumor resistance (PMID: 33076450).
The authors should highlight their findings and suggest how their findings can be used in the clinical setting and how can they be applied to designing possible treatments for cancer.
Please correct some minor typos throughout the text.
Reviewer 3 Report
The manuscript by Lee et al. investigated the lncRNA GAS5 regulation of colon cancer cell response to 5-fluorouracil (5-FU) treatment. The authors showed that GAS5 expression is lowered in 5-FU resistant cells, and modulation of GAS5 levels influenced proliferation and viability of HCT116 cells. They further confirmed that GAS5 associates with BAX mRNA, and that miR-128-3p may compete with GAS5 for BAX mRNA binding to control BAX protein expression. They postulated that in response to 5-FU treatment, GAS5 promotes cell death via increasing pro-apoptotic BAX protein expression, and overexpression of GAS5 sensitizes anti-cancer drug response in resistant cells. Overall, these findings are interesting however, the authors need to improve the quality of the experimental data to further support their claims in the manuscript prior to publication.
Specific points that need to be addressed:
1. The authors showed decrease in expression of lncRNA GAS5 and Bax protein in 5-FU resistant versus parental colon cancer cell lines. What are the levels of miR-128-3p in these cells?
2. The quantification of the G1 cell cycle phase illustrated in Figures 2C should also include the other cell cycle phases.
3. The sensitization of 5-FU resistant cells to anticancer drugs via overexpression of GAS5 presented in Figure 6C is weak and vague. These experiments need to be repeated for clarification. Especially that, based on the large error bars and small differences in cell viability, the statistical significance of the data is concerning.
4. Description of Figure 1 is unclear and needs to be corrected. The authors mentioned there Table 4 that is not present in the manuscript.
5. The manuscript needs some revision in spelling and grammar.
Round 2
Reviewer 3 Report
The authors have sufficiently addressed the prior concerns and improved the manuscript. Thus, the paper is suitable for publication.